# Macroecological patterns in experimental microbial communities

**William R. Shoemaker** [ID][1]*, **Álvaro Sánchez**[2], **Jacopo Grilli**[1]

**1** Quantitative Life Sciences, The Abdus Salam International Centre for Theoretical Physics (ICTP), Trieste, Italy, **2** Instituto de Biología Funcional y Genómica, IBFG-CSIC, Universidad de Salamanca, Salamanca, Spain

* williamrshoemaker@gmail.com

**Data availability statement:** All code written for this study is available on GitHub under a Creative Commons Attribution 4.0 International License: https://github.com/wrshoemaker/experimental_macroecology. Raw data are

## Abstract

Ecology has historically benefited from the characterization of statistical patterns of biodiversity within and across communities, an approach known as macroecology. Within microbial ecology, macroecological approaches have identified universal patterns of diversity and abundance that can be captured by effective models. Experimentation has simultaneously played a crucial role, as the advent of high-replication community time-series has allowed researchers to investigate underlying ecological forces. However, there remains a gap between experiments performed in the laboratory and macroecological patterns documented in natural systems, as we do not know whether these patterns can be recapitulated in the lab and whether experimental manipulations produce macroecological effects. This work aims at bridging the gap between experimental ecology and macroecology. Using high-replication time-series, we demonstrate that microbial macroecological patterns observed in nature exist in a laboratory setting, despite controlled conditions, and can be unified under the Stochastic Logistic Model of growth (SLM). We found that demographic manipulations (e.g., migration) impact observed macroecological patterns. By modifying the SLM to incorporate said manipulations alongside experimental details (e.g., sampling), we obtain predictions that are consistent with macroecological outcomes. By combining high-replication experiments with ecological models, microbial macroecology can be viewed as a predictive discipline.

## Author summary

Determining whether an empirical pattern can be manipulated is a crucial step towards building a predictive theory. Our study aimed to determine the extent that experimental manipulation could recapitulate ecological patterns observed in natural microbial communities with minimal fine-tuning. Specifically, we investigated the macroecology (i.e., statistical ecological patterns) of an experiment where a large number of microbial communities were maintained over time. We found that many of the macroecological patterns observed in nature can also be found in experiments. Such patterns can

available on the NCBI SRA database under accession numbers PRJNA761777 https://www.ncbi.nlm.nih.gov/bioproject/PRJNA761777 and PRJNA761387 https://www.ncbi.nlm.nih.gov/bioproject/?term=PRJNA761387. Processed data for this project is available on Zenodo (DOI: 10.5281/zenodo.8393848).

**Funding:** This work was supported by the NSF Postdoctoral Research Fellowships in Biology Program under Grant No. 2010885 (W.R.S.), Fondo Italiano per la Scienza - FIS (CUP J53C23002290001; J.G.), and Grant PID2021-125478NA-I00 funded by CIN/AEI/10.13039/501100011033 and by "ERDF A way of making Europe" (A.S.). The funders had no role in study design, data collection and analysis, decision to publish, or preparation of the manuscript.

**Competing interests:** The authors have declared that no competing interests exist.

be altered by manipulating the ecological force of migration. A minimal model of ecological dynamics that incorporated experimental details (e.g., sampling) was able to capture the consequences of these ecological manipulations. The results of this work establish a demarcation between patterns that are and are not surprising given our understanding of microbial ecological dynamics and demonstrate the potential for microbial macroecology as a predictive discipline.

## Introduction

Microbial communities inhabit virtually every environment on Earth. Through their ubiquity, abundance, and diversity, microorganisms regulate the biogeochemical processes that sustain life on the biosphere. Furthermore, host-associated microbial communities play a crucial role in maintaining the health of many macroscopic forms of life, including humans, whose health is impacted by their microbiome [1,2]. Given their environmental, medical, and economic importance, it is necessary to develop quantitative theories of ecology that allow researchers to explain, maintain, and alter the properties of microbial communities. A challenge of this scale is daunting, and the complexity of microbial communities has promoted the engagement of researchers from various disciplines with distinct approaches. Of these approaches, there are two that have substantially contributed towards our quantitative understanding of microbial communities in recent years: macroecology and experimental ecology.

Historically, the field of ecology has achieved considerable success by characterizing generalized patterns of biodiversity, an approach known as macroecology [3–9]. The macroecological approach is, fundamentally, statistical in nature, allowing for quantitative predictions and extrapolations to be made about the typical features of ecological communities without having to specify microscopic ecological forces. The generalized nature of this approach has allowed researchers to successfully characterize a diverse array of microbial ecological patterns [10–19] and spurred the development of mathematical models of microbial ecology grounded in statistical physics [20–28].

Through the macroecological approach, disparate patterns were recently unified by the observation that the typical microbial community follows three macroecological patterns: 1) the abundance of a given community member across communities follows a gamma distribution, 2) the mean abundance of a given community member is not independent of its variance (i.e., Taylor's Law [29–31]), and 3) the mean abundance of a community member across communities follows a lognormal distribution [32]. These three general patterns can be captured by an intuitive mathematical model of density-dependent growth with environmental noise, the stochastic logistic model (SLM) of growth [32–34]. Building on its utility, the SLM has been successfully extended to quantitatively capture additional empirical microbial macroecological patterns. Examples that explicitly use the SLM include attempts to capture measures of ecological distances and dissimilarities between communities [35], alternative stable-states [36], correlations in abundance between community members [37], community-level measures across taxonomic and phylogenetic scales [38], and dynamics within and across human hosts at the sub-species level (i.e., strains) [18,39]. The results of these studies demonstrate that a minimal mathematical model of ecological dynamics can capture a broad assemblage of microbial macroecological patterns.

The generality of macroecology is a useful feature that permits the construction of a working theory of microbial ecology [40,41]. However, there is a trade-off between this

generality and the ability of macroecology to provide causal connections between ecological mechanisms and observed patterns (e.g., the exponent of Taylor's Law) [42]. This lack of causality is due to the inherent difficulty of manipulating macroecological patterns in natural communities. This study addresses this issue by expanding the scope of the macroecological approach from natural systems to experimental communities, providing the means to investigate how mechanistic manipulations alter macroecological patterns.

Experimentation has played a crucial role in the documentation and manipulation of ecological forces [43,44]. The advent of 16S rRNA amplicon sequencing allows researchers to investigate the ecological dynamics of a large number of replicate microbial communities in a laboratory setting. This controlled approach has proven to be highly successful, allowing researchers to determine the extent that the assembly of microbial communities is reproducible and predictable [45–52]. Surprisingly, seemingly simple environments can maintain highly dissimilar microbial communities, even in systems harboring a single exchangeable resource [53]. This observation has led researchers to investigate the susceptibility of microbial communities to experimentally imposed ecological forces. A prominent example is the migration of individuals between communities, an ecological force that is amenable to experimental manipulation [54,55] and can alter the heterogeneity of communities across replicates [53,56,57].

In this study we sought to bridge the gap between microbial macroecological patterns observed in nature and experiments performed in the lab. We consider experimentally assembled microbial communities that were exposed to qualitatively different forms of migration [53]. The original goals of this experiment were to evaluate the functional and taxonomic convergence of a large number ($\sim 100$) of replicate microbial communities in environments supplied with a single carbon source (i.e., glucose). This experiment provides sufficient data to investigate macroecological patterns due to the authors' decision to maintain a large number of replicate communities for a given treatment, to sequence a moderate number of communities over time, and to have periods of time where migration does and does not occur within a given community. Using this experiment, we first demonstrate that macroecological patterns observed in natural communities can be recapitulated in experimentally assembled communities. By connecting these patterns to the predictions of the SLM, we developed a model of microbial macroecology that incorporates experimental details. Specifically, we focus on two migration treatments. The first treatment (referred to as regional migration) corresponds to a classical mainland-island scenario [58], where migrants from the community from which replicate communities were originally assembled (known as the progenitor community) continued to migrate over time. The second case, referred here as global migration, corresponds to a fully-connected metacommunity model [59], where migration occurred between communities that were assembled from the same progenitor community. We examine the ways in which the ecological force of migration can be manipulated, their resulting macroecological outcomes, and the degree that these outcomes can be predicted by the SLM. Using these results, we identify when and how the SLM succeeded in predicting the effects of experimental manipulation. By leveraging high-throughput ecological experiments alongside robust statistical patterns, we can strengthen the predictive and quantitative elements of microbial ecological theory.

## Materials and methods

### Experimental data

We identified an appropriate dataset to investigate the macroecology of experimental microbial communities. A recent experimental study maintained $\sim 100$ replicate communities

assembled from a single progenitor soil community and imposed a variety of demographic treatments [53]. Here, a given microcosm was inoculated from a single progenitor community. All microcosms were provided with glucose as the sole supplied carbon source. Each community was allowed to grow for 48 hours. A fraction of the volume $D_{\text{transfer}} = 0.008$ (i.e., an aliquot ratio of 1:125) was then sampled and used to inoculate a new microcosm with the same profile of supplied resources (Table 1). This procedure of inoculating a microcosm with an aliquot and allowing it to grow for a given period of time is known as a transfer cycle (Fig 1a). Transfer cycles were repeated 18 times per replicate community.

The demography of replicate communities was systematically altered by manipulating the form of migration. For the first manipulation, aliquots of the progenitor community were mixed with aliquots from the previous transfer cycle, so that the community composition of a microcosm at the start of a new transfer cycle was comprised of both the composition of the progenitor and the previous transfer cycle. This manipulation is referred to as regional migration. In the second case, migration was manipulated by sampling aliquots of each community at the end of a given transfer cycle. These aliquots were mixed and redistributed at the start of the subsequent transfer cycle at a dilution rate $\approx 0.0084$ (Table 1; dilution ratio of 504:60,000). This manipulation is referred to as global migration. Migration manipulations were only performed for the first 12 dilution cycles (out of the total 18 transfer cycles). This experimental design provides a series of time points where migration was and was not performed within each replicate community.

A number of communities were also inoculated with a large aliquot of the progenitor community. Macroecological patterns of these communities were examined but were excluded from subsequent analyses since this manipulation only occurred at the first transfer cycle and did not induce lasting macroecological effects. Community data for this experiment was obtained via 16S rRNA sequencing. We reprocessed all raw FASTQ data from the original study to obtain Amplicon Sequence Variants (ASVs) using the package DADA2 so that read counts of one could be inferred by pooling observations to obtain a reliable estimate of their error rate [60]. ASVs differ from the more typical Operational Taxonomic Units in that they allow for community members to be resolved down to single-nucleotide differences rather than relying on clustering sequences by a given similarity (see [61,62]). All replicate communities were sequenced at transfers 12 and 18. A subset of communities were sequenced at every transfer. The number of replicate communities sequenced at a given transfer cycle for a given treatment is summarized in S1 Table. S1 Text summarizes technical details about the experiment, which can also be found in the original manuscript [53].

**Table 1. Experimentally imposed parameters that were used in this study [53]. Estimates of total community size were previously obtained. The calculation of $N_{\text{global}}$ is the result the procedure used to pool samples from replicate communities into a single global pool at the end of each transfer, which was then used to inoculate communities at the start of the subsequent transfer.**

| Quantity | Symbol | Value |
|---|---|---|
| Total abundance, transfer | $N^{*}(T)$ | $\sim 10^{8}$ |
| Total # of migrants, regional | $N_{\text{regional}}$ | $7.92 \cdot 10^{6}$ |
| Total # of migrants, global | $N_{\text{global}}$ | $M \cdot (8 \cdot 10^{-5}) \cdot D_{\text{transfer}} \cdot N^{*} \approx 3.07 \cdot 10^{4}$ |
| Dilution rate, transfer | $D_{\text{transfer}}$ | $\frac{4\mu\text{L}}{500\mu\text{L}} = 0.008$ |
| Dilution rate, global | $D_{\text{global}}$ | $\frac{504\mu\text{L}}{60,000\mu\text{L}} \approx 0.0084$ |
| Dilution rate, regional | $D_{\text{regional}}$ | $\frac{504\mu\text{L}}{60,000\mu\text{L}} \approx 0.0084$ |

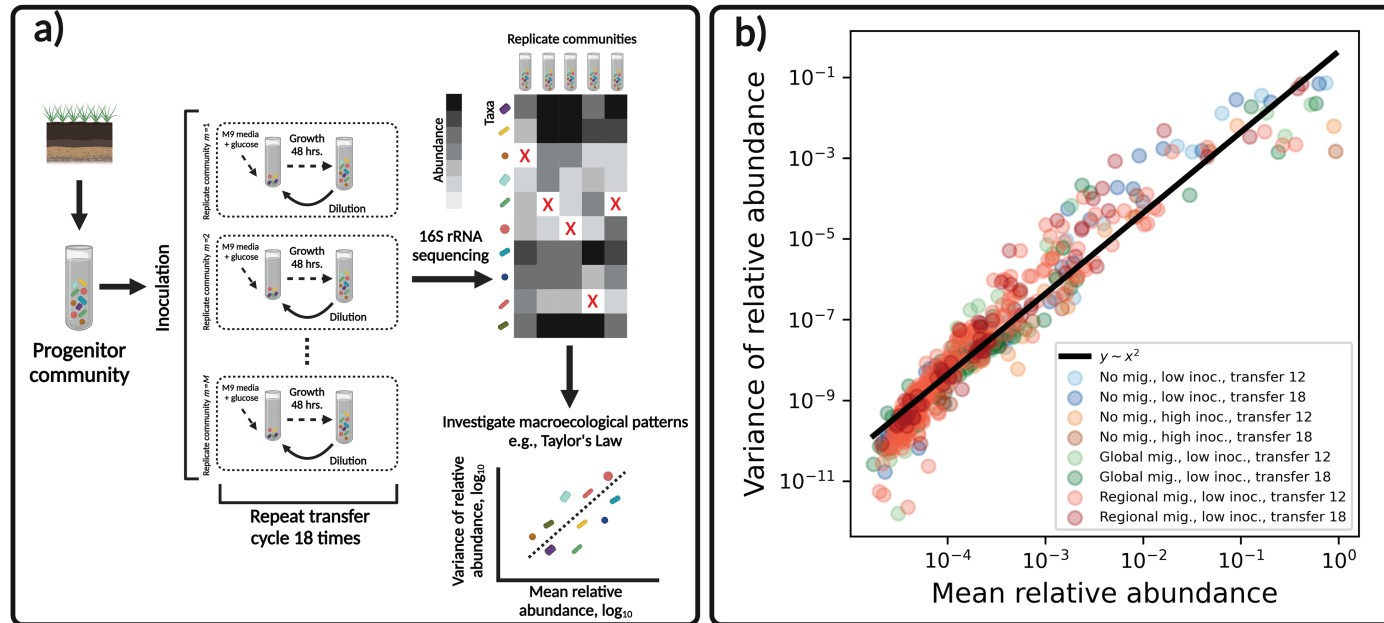

**Fig 1. Emergent variation in experimental communities. a)** Microbial community assembly experiments provide a means to maintain diversity in a laboratory setting. These experiments are commonly performed by growing a community sampled from a given environment (e.g., soil) and using it to inoculate a large number of replicate microcosms containing the same resource (e.g., M9 minimal media with glucose as the sole carbon source). These replicate communities are allowed to grow for a period of time (e.g., 48 h.) before aliquots are taken and diluted into microcosms containing replenished resource. This process known as a "transfer cycle" is repeated a given number of times (e.g., 18) and the resulting communities are sequenced via 16S rRNA sequencing. The abundances of community members have the potential to provide the variation necessary to investigate the existence of macroecological patterns in an experimental setting (e.g., Taylor's Law), a goal of this study. Red "X" symbols represent the absence of a community member in a sample. **b)** Variation in abundance consistently arose across treatments and timescales in experimental communities. This variation could be captured by the relationship between the mean and variance of relative abundance, a pattern known as Taylor's Law. Each data point represents statistical moments calculated across replicate communities for a single ASV from a single experimental treatment. Fig 1a created with Biorender (ID a78p433).

## The macroecological predictions of the stochastic logistic model

Populations at low abundance often grow at an initially high rate that proceeds to decrease as they approach the maximum abundance that an environment can support. This is the central idea underlying logistic growth and it can be captured by two parameters: 1) the minimal length of time required to reproduce (inverse of the maximum growth rate), $\tau$ and 2) the carrying capacity of an environment, $K$. This simple picture assumes a fully deterministic environment. Environmental stochasticity can be added as fluctuations in the growth rate, where the strength of environmental noise is controlled by the parameter $\sigma$, representing the coefficient of variation of growth rate fluctuations. The Stochastic Logistic Model (SLM) captures these two fundamental ecological processes under the form of a Langevin equation [63,64]

$$\frac{dx_i}{dt} = \underbrace{\frac{x_i}{\tau_i}\left(1 - \frac{x_i}{K_i}\right)}_{\text{Self-limiting growth}} + \underbrace{\sqrt{\frac{\sigma_i}{\tau_i}}x_i \cdot \eta_i(t)}_{\text{Environmental noise}} \tag{1}$$

where $x_i$ is the relative abundance of ASV $i$. The term $\eta_i(t)$ introduces stochasticity into the equation as white noise, where the expected value of is $\langle \eta(t) \rangle = 0$ and the time correlation is a delta function $\langle \eta(t)\eta(t') \rangle = \delta(t - t')$ (implying that the noise at time $t$ is

uncorrelated with the noise at time $t'$). This model can be extended to time-correlated environmental fluctuations [32].

The SLM captures three empirical macroecological patterns that are found in microbial communities across natural environments: 1) gamma distributed abundances, 2) a power law relationship between mean relative abundance and the variance of relative abundance (i.e., Taylor's Law), and 3) a lognormally distributed distribution of mean relative abundances [32]. To provide the necessary context for the study, we briefly summarize these three patterns and their connection to the SLM.

**The abundance fluctuation distribution.** The Langevin equation defined in Eq 1) defines a stochastic trajectory $x_i(t)$. The probability $P(x_i, t | x_i(0), 0)$ that the abundance of species $i$ equals $x_i(t)$ at time $t$ when starting from initial condition $x_i(0)$ at time $t = 0$ can be obtained from the Fokker-Planck equation corresponding to Eq 1 [63]. For large times, this probability converges to the stationary probability distribution $P^*(x_i)$, which in the case the distribution of the SLM takes the form of a gamma distribution

$$P^*(x_i | \bar{x}_i, \beta_i) = \frac{1}{\Gamma(\beta_i)} \left( \frac{\beta_i}{\bar{x}_i} \right)^{\beta_i} \exp\left[ -x_i \frac{\beta_i}{\bar{x}_i} \right] x_i^{\beta_i - 1} \tag{2}$$

where the two parameters $\bar{x}_i$ and $\beta_i$ can be expressed in terms of the statistical moments of the distribution and also in terms of parameters appearing in Eq 1

$$\langle x_i \rangle = \bar{x}_i = K_i \left( 1 - \frac{\sigma_i}{2} \right) \tag{3a}$$

$$\frac{\text{Var}(x)}{\langle x_i \rangle^2} = \frac{\sigma_i}{2 - \sigma_i} = \frac{1}{\beta_i} \tag{3b}$$

The expression above hold for absolute abundances, while we must content with a finite number of reads obtained by sequencing, which are affected by both compositionality and sampling effects. We can model our sampling process as the Poisson limit of a multinomial distribution, obtaining a form of the AFD that explicitly accounts for the effect of sampling [32]. Using this distribution, we can calculate the probability of obtaining $n$ reads out of a total sampling depth $N$ for the $i$th ASV as

$$P(n_i | \bar{x}_i, \beta_i, N) = \frac{\Gamma(\beta_i + n_i)}{n_i! \Gamma(\beta_i)} \left( \frac{\bar{x}_i N}{\beta_i + \bar{x}_i N} \right)^{n_i} \left( \frac{\beta_i}{\beta_i + \bar{x}_i N} \right)^{\beta_i}. \tag{4}$$

Reprocessing sequence data using `DADA2` was necessary for read counts of $n = 1$ to be inferred, allowing us to investigate the full distribution of $P(n_i | \bar{x}_i, \beta_i, N)$. Often in ecology we are interested in the relationship between the mean relative abundance of a community member and the fraction of communities where it is present (i.e., its occupancy) [13,65,66]. We can derive a prediction of occupancy using the sampling form of the AFD by setting $n = 0$ and noticing that probability that an ASV is present is the complement of its absence. Averaging over $M$ communities, one obtains a prediction of occupancy

$$\langle o_i \rangle = 1 - \frac{1}{M} \sum_{m=1}^{M} P(0 | \bar{x}_i, \beta_i, N_m). \tag{5}$$

Reprocessing the sequence data so that singletons could be reliably inferred allows us to sum from 1,...,$M$ rather than 2,...,$M$. Prediction error was estimated as described in S2 Text.

It is worth stating that a stationary probability distribution does not necessarily correspond with a single stochastic differential equation. In the case of microbial abundances, an alternative model has been proposed that results in a stationary gamma distribution, but where stochasticity arises due to demographic fluctuations (i.e., proportional to $\sqrt{x_i}$ rather than $x_i$) [16]. This model is mathematically the same as a migration-birth-death model that has been examined elsewhere [32,67]. Under the migration-birth-death interpretation, the absence of migration (as implemented in our no migration treatment) would result in the abundance of an observed species increasing over time without bound, a prediction that is not borne out by the data. Given this prediction alongside prior evidence that environmental noise succeeds in explaining temporal macroecological patterns of natural communities, we elected to incorporate stochasticity as environmental noise [34].

**Taylor's Law.** Taylor's Law describes the empirical relationship between the mean and variance of the relative abundance [29,30] across ASVs

$$\mathrm{Var}(x_i) \propto \langle x_i \rangle^{2b} . \tag{6}$$

One can generate a prediction for the dependency of the exponent $b$ on ecological parameters by deriving the mean and variance for their probability distribution and plugging them into the above expression. By comparing this expression to Eq 3 one can see that the case where $\beta_i^{-1}$ is independent of $\bar{x}_i$ corresponds to $b = 1$. In the context of the SLM, this choice implies that the strength of environmental noise is constant for all ASVs ($\sigma_i = \sigma$). However, for batch culture cycles we are fundamentally interested in how $b$ depends on the initial abundance of a species, which in-turn depends by migration. Investigating this dependency requires the time-dependent solution of the SLM which, while solved, is rather unwieldy (S5 Text). Therefore, we elected to perform simulations of the SLM to determine how the exponent responds to ecological parameters (additional information in the the following section). Estimates of $b$ were obtained by log transforming both axes and performing ordinary least squares regression with `SciPy v1.4.1`.

**The lognormally distributed mean abundance distribution.** In observed data, the mean relative abundances (calculated either across communities or over time) of ASVs, known as the Mean Abundance Distribution (MAD), frequently follows a lognormal probability distribution

$$P(\bar{x}_i)^{\mathrm{lognorm}} = \frac{1}{\sqrt{2\pi s^2}\bar{x}_i} \exp\left[-\frac{(\ln \bar{x}_i - \mu)^2}{2s^2}\right] \tag{7}$$

If we were to apply this distribution to investigate data it would be the equivalent to assuming perfect sampling. In reality, we cannot rule out *a priori* the existence of community members with mean abundances that lie below the limit of detection set by sampling (i.e., # of reads). To account for sampling effects, we use a modified form of the lognormal that considers mean abundances with read counts greater than

$$P(\bar{x}_i)^{\mathrm{lognorm}}_{\mathrm{emp}} = \frac{\theta(\bar{x}_i - c)P(\bar{x}_i)^{\mathrm{lognorm}}}{\int \theta(z - c)P(z)^{\mathrm{lognorm}}dz} \tag{8}$$

where $\theta(\cdot)$ is the Heaviside step function. Parameters of this modified lognormal were fit to the empirical MAD as previously described (see Supplementary Note 7 in [32]).

### Incorporating migration into the SLM

The original form of the SLM is not directly applicable to microbial community assembly experiments. As discussed above, such experiments are performed as transfer cycles, where growth occurs over a period of time within a microcosm before an aliquot is transferred to an environment with replenished resources. Phenomenological models of logistic growth are appropriate for modeling dynamics within a single transfer cycle under certain conditions (S3 Text). In addition, the time-dependent form of Eq 9 has been derived (i.e., $P(x_i(t), t|x_i(0), 0)$ and could be used to model the temporal dynamics within a given transfer cycle [68] (description of solution in S5 Text). However, no attempt has been made to incorporate experimental details into the SLM so that it can be used to obtain predictions for the macroecological effects of experimental manipulations.

First, we extended the SLM to incorporate the transfer cycle process. We consider a piecewise form of the SLM to describe the dynamics within the $k$th transfer cycle

$$\frac{dx_i^{(k)}}{dt} = \frac{x_i^{(k)}}{\tau_i}\left(1 - \frac{x_i^{(k)}}{K_i}\right) + \sqrt{\frac{\sigma_i}{\tau_i}}x_i^{(k)} \cdot \eta_i(t) \tag{9}$$

where the dynamics start at $t = 0$ and continue until time $t = T$. At time $T$ an aliquot of the community is sampled. The volume of the aliquot relative to the total volume of the microcosm is known as the transfer dilution rate $D_{\text{transfer}}$. If the total abundance of the community is known at time $T$, then the total abundance at the start of the next transfer cycle can be calculated as $N^{(k)}(0) = D_{\text{transfer}} \cdot N^{(k-1)}(T)$. The process of sampling community members at the end of a transfer cycle can be modeled as multinomial sampling process (S6 Text). With this sampling process and our piece-wise form of the SLM, we have a minimal dynamical model of a microbial community assembly experiment.

While they represent different forms of migration, both treatments are similar in that they are implemented by altering the initial abundance of a community member at the start of a transfer cycle. We can formulate the number of cells sampled due to migration at the start of a transfer cycle for a given ASV as follows

$$n_{i,\text{regional}}(0)^{(k)} \propto x_{i,\text{progenitor}} \tag{10a}$$

$$n_{i,\text{global}}(0)^{(k)} \propto \frac{\sum_{m=1}^{M} n_{i,m}^{(k-1)}(T)}{\sum_{i=1}^{S}\sum_{m=1}^{M} n_{i,m}^{(k-1)}(T)} \tag{10b}$$

The sampling process of migrants can again be modeled using a multinomial distribution (S6 Text). We can now examine how the mean initial *relative* abundance of a given ASV depends on migration. In this experiment the total abundance of a community at the end of a transfer cycle did not considerably vary from transfer to transfer, meaning that we can assume that the total abundance at time $T$ within a transfer cycle remained the same for all transfer cycles $k$, $(N^{(k-1)}(T) \approx N^{(k)}(T) \equiv N^*(T))$. Using this result, we divide the abundance of each ASV by the total abundance in a community to obtain relative abundances $(x_i^{(k-1)}(T) = \frac{n_i^{(k-1)}(T)}{N^*(T)})$, obtaining the following prediction for the mean abundance at the start of a transfer cycle

$$\left\langle x_i^{(k)}(0) \right\rangle = \begin{cases} \left\langle \dfrac{D_{\text{transfer}} x_i^{(k-1)}(T) N^*(T) + D_{\text{regional}} x_{i,\text{regional}} N_{\text{regional}}}{D_{\text{transfer}} N^*(T) + D_{\text{regional}} N_{\text{regional}}} \right\rangle, & \text{Regional} \\[2ex] \left\langle \dfrac{D_{\text{transfer}} x_i^{(k-1)}(T) N^*(T) + D_{\text{global}} x_{i,\text{global}}^{(k-1)} N_{\text{global}}}{D_{\text{transfer}} N^*(T) + D_{\text{global}} N_{\text{global}}} \right\rangle, & \text{Global} \\[2ex] \left\langle x_i^{(k-1)}(T) \right\rangle, & \text{No migration} \end{cases} \tag{11}$$

The impact that migration manipulations have on the initial relative abundance of the AFD and how they compare to the typical view of migration as a process that occurs at a constant rate is illustrated in S3 Fig. Parameter values are calculated using experimental details and can be found in Table 1 and simulation details of Eq 9 can be found in S6 Text.

### Treatment-specific migration statistics

We identified appropriate statistics to capture macroecological outcomes specific to the regional and global migration treatments. Significance was established in all instances using null distributions of statistics obtained via permutation.

**Regional migration statistics.** We sought to identify 1) how the correlation in $\langle x_i(t) \rangle$ between no migration and migration treatments increased after the cessation of migration and 2) how the dependency of $\langle x_i(t) \rangle$ on $x_{i,\text{progenitor}}$ dissipated once migration manipulations ceased. The change in correlation coefficients ($\rho$) between transfers 12 and 18 was assessed using Fisher's $Z$ statistic [69]

$$Z_\rho = \frac{z_{18} - z_{12}}{\sqrt{\frac{1}{M_{12}-3} + \frac{1}{M_{18}-3}}} \tag{12}$$

where $z_t = \log\left[\frac{1+\rho_t}{1-\rho_t}\right]$ and $M_k$ is the number of replicate communities at transfer $k$. The denominator represents the standard error of the numerator. Dependency on the progenitor was evaluated by fitting a regression between $x_{i,\text{progenitor}}$ and the ratio of mean abundances in the regional and no migration treatments $\frac{\langle x_i \rangle_{\text{regional}}}{\langle x_i \rangle_{\text{no mig}}}$. Regression fits were obtained at transfers 12 and 18 and the statistical significance of the change in slope was evaluated using a permutation-based $t$-test.

**Global migration statistics.** For the global migration treatment we predicted that fluctuations in abundance would decrease under global migration. We performed two types of analyses to test our fluctuation predictions: examining whether 1) the correlation in $\text{CV}_{x_i}$ between no migration and global migration treatments and 2) the CV of the log-ratio of relative abundances between consecutive transfers increased after the cessation of migration. The log-ratio of relative abundances between two timepoints is defined as $\Delta\ell_i^{(k)} = \log\left[x_i^{(k+1)}/x_i^k\right]$.

We elected to use $\Delta\ell_i^{(k)}$ because 1) it can be interpreted as a discretized form of the per-capita growth rate and 2) as a ratio it cancels out any potential multiplicative time-independent sample biases [70,71]. We first calculated the CV of $\Delta\ell_i^{(k)}$ across communities at each transfer for each ASV, $\text{CV}_{\Delta\ell_i}^{(k)}$. To contrast this measure of fluctuations *over replicates* with a measure of fluctuations over time *within a replicate*, we calculated the CV before and after the cessation of migration *for each replicate*, $\text{CV}_{\Delta\ell_{i,m}}^<$ and $\text{CV}_{\Delta\ell_{i,m}}^>$. The effect of the cessation of migration on a per-replicate basis was examined using an $F$ statistic designed to compare two CVs [72]

$$F_{\text{CV}}^{(i,m)} = \frac{(\text{CV}_{\Delta\ell_{i,m}}^>)^2}{1 + (\text{CV}_{\Delta\ell_{i,m}}^>)^2 \frac{M^>-1}{M^>}} \cdot \frac{1 + (\text{CV}_{\Delta\ell_{i,m}}^<)^2 \frac{M^<-1}{M^<}}{(\text{CV}_{\Delta\ell_{i,m}}^<)^2} \tag{13}$$

where $M^<$ and $M^>$ represent the number of transfers before and after the cessation of migration, respectively.

### Evaluating SLM predictions

Using our experiment-informed simulation, we determined whether shifts in macroecological quantities due to migration could be explained by the SLM (S6 Text). To briefly summarize, we first simulated the experiment for $10^4$ uniformly drawn sets of $\{\tau, \sigma\}$. We used Approximate Bayesian Computation (ABC) to identify the parameters where the simulated macroecological pattern had the lowest Euclidean distance to the observed pattern. We then used this inferred set of parameters to generate distributions of the macroecological pattern or summary statistic using $10^3$ SLM simulations. To evaluate whether migration was capable of altering a pattern under the SLM, we repeated our simulations across a grid of $\{\tau, \sigma\}$ combinations.

## Results

Using a high-replication community assembly experiment, we first examined whether universal macroecological patterns observed in natural communities could be recapitulated in a laboratory setting. We then examined whether the patterns we observed were susceptible to experimental manipulation. We rationalized our observations by extending the SLM to incorporate experimental details, allowing us to balance the effectiveness of the SLM as a minimal model of ecology with the need for experimental realism. Using simulations, we obtained quantitative predictions and identified whether the macroecological consequences of a given form of migration could be captured by the SLM, establishing its capacity to predict patterns of microbial biodiversity.

### Macroecological patterns emerge in experimental systems

Predicting the outcomes of experimental manipulation is a vital goal of microbial ecology. With this goal in mind, we first determined the degree to which empirical macroecological patterns documented in observational data held in an experimental system (Fig 1a). The number of potential patterns one can examine is potentially large. From previous works on natural microbial communities, we know that three main patters are sufficient to recapitulate several others [32]: 1) the mean and variance of species abundances displaying non-independence (Taylor's Law), 2) the Abundance Fluctuation Distribution across independent sites (AFD) following a gamma distribution, and 3) the Mean Abundance Distribution across independent sites (MAD) following a lognormal distribution. Other commonly studied macroecological patterns (e.g., the Species Abundance Distribution, abundance-occupancy relationship), can be obtained and predicted as a consequence of these three patterns. Furthermore, these three patterns can be rationalized by a stochastic mean-field model, the Stochastic Logistic Model of growth [32].

 We first quantified the variation in abundance that emerged in experimental communities. The average relative abundance varied over four orders of magnitude, meaning that a considerable degree of community-level variation can be maintained in a laboratory setting. We found that the shape of the relationship between the mean and variance of relative abundance across ASVs did not qualitatively vary across treatments, implying that Taylor's Law can be applied [29]. Fig 1b shows that the variance of relative abundance was proportional to the square of the mean, corresponding to Taylor's Law with an exponent equal to two. This value implies that the coefficient of variation of relative abundances (CV) remained constant across

ASVs. Fitting the exponent to each treatment for each transfer, we found a mean exponent of $2.1 \pm 0.06$, suggesting that despite the variation in typical abundance, the CV of relative abundances remained roughly constant across ASVs.

Given the existence of substantial fluctuations of abundance across replicate communities, we focused on the full distribution of abundances, known as the Abundance Fluctuation Distribution (AFD). To facilitate comparisons across ASVs and treatments, we rescaled the logarithm of the AFD for each ASV by its mean and variance (i.e., the standard score). We found that rescaled AFDs from different treatments tended to collapse on a single curve, implying that despite differences in experimental details, the general shape of the AFD remained invariant. Fig 2a shows that the bulk of empirical AFDs from different treatments generally followed a gamma distribution, as predicted by the SLM (Eq 2). High abundances tended to fall outside the predicted right tail of the gamma distribution, a deviation we attributed to the existence of alternative stable states (and will revisit later) [53].

Abundance fluctuations offer only a partial view of community variation since patterns of presence/absence of community members could display non-trivial patterns. We therefore studied the fraction of communities where a given ASV was present, a measure known as occupancy. Given that the process of sampling ASVs within a community can be modeled as a Poisson process, the distribution of read counts can be derived from a gamma AFD which can be used to obtain occupancy predictions (Eq 5). We found that the predictions of the SLM generally held across treatments, with slight deviations at high values of observed occupancies (S1 Fig). This trend is reflected in the distribution of relative errors in our occupancy predictions, where certain treatments appeared to have higher error values than others (S1 Fig). A permutation-based test was performed to establish the statistical significance of this observation (S1 Fig). We found a significant, though slight, effect of migration, where it reduced the error of our predictions for both regional ($\overline{\Delta\varepsilon}_{\text{regional}} = -0.24$, $P = 0.001$) and global migration ($\overline{\Delta\varepsilon}_{\text{global}} = -0.2$, $P = 0.01$). Regardless, Eq 5 predicted the occupancy of the typical ASV across treatments with a high level of accuracy.

Leveraging this result, we investigate the relationship between the mean relative abundance across replicates and the occupancy of an ASV, more commonly known as the abundance-occupancy relationship [13,65,66]. We found that all treatments followed predictions of a gamma distributed AFD (Fig 2b). Deviations from the prediction were likely driven by a combination of 1) averaging over a finite number of ASVs and 2) variation of $\beta_i$ across ASVs [36]. The existence of a relationship between average abundance and occupancy in experimental communities is particularly striking, as it implies that the probability of observing a given community member is primarily determined by sampling effort (i.e., # reads) and mean relative abundance, despite differences in experimental details. As a point of comparison, we note that prior research efforts into the AFD of natural communities found that while the gamma distribution succeeded in capturing the shape of the AFD and predicting occupancy, alternative distributions such as the lognormal failed on both counts [32].

The success of the gamma distribution in predicting occupancy and interpreting Taylor's Law implies that differences in abundances are primarily driven by differences in their mean relative abundance. The distribution of the mean relative abundances of taxa (known as Mean Abundance Distribution, MAD) in data is well-described by a lognormal distribution ([32]; Eq 8). This observation can be seen as an across-community extension of the observation that the distribution of abundances within a single microbial community could be captured by a lognormal distribution [11]. The low richness of experimental communities ($\sim 50$ ASVs per community) makes it difficult to study the MAD of a given treatment, requiring ASVs to be pooled across treatments and transfers. Using a form of the lognormal that accounts

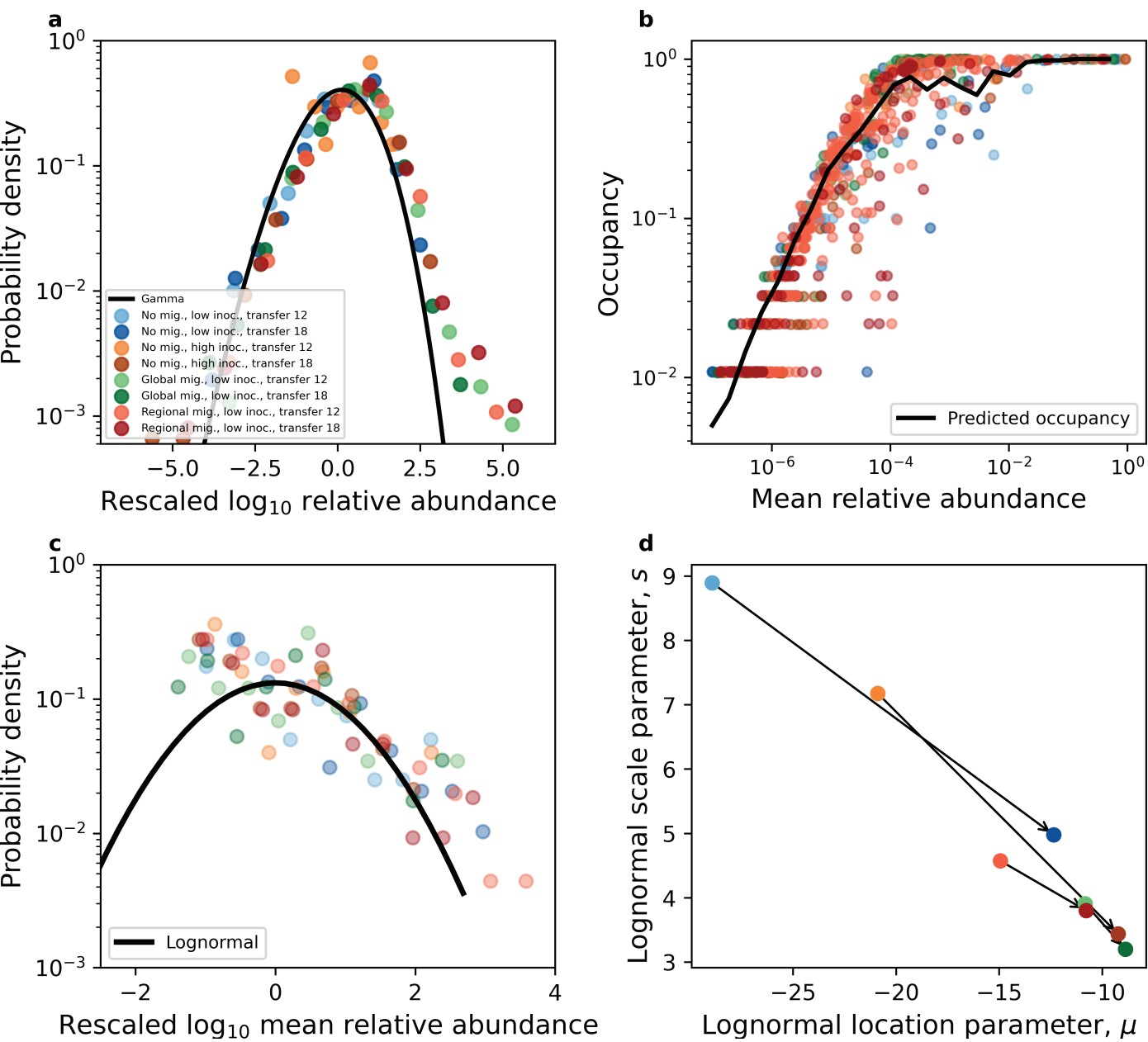

**Fig 2. Macroecological patterns hold in experimental communities.** Empirical macroecological patterns that were previously identified in natural microbial systems consistently arise in experimental communities [32]. **a)** The Abundance Fluctuation Distribution (AFD) tended to follow a gamma distribution across treatments. **b)** A gamma distribution that explicitly considers sampling successfully predicted the fraction of communities where an ASV is present (i.e., its occupancy). The prediction of the gamma (black line) was obtained by averaging over ASVs within a given mean relative abundance bin. **c)** The Mean Abundance Distribution (MAD) was similar across treatments and largely follow a lognormal distribution. The left side of the distribution represents community members with mean abundances below the detection limit set by finite sampling (Materials and Methods; [32])**d).** By examining the location parameter ($\mu$) and the scale parameter ($s$) of the lognormal distribution for each treatment, we found that these independently inferred parameters converged towards the same value as the experiment progressed from transfer 12 (light shade) to transfer 18 (dark shade). These two parameters control the shape of the MAD, meaning that different treatments are converging to similar distributions of mean abundance.

for sampling effort (Eq 8), we found that the resulting empirical MAD can be captured by a lognormal (Fig 2c).

It is important to note that the lognormality of the MAD itself does not validate or invalidate the SLM. Rather, under the SLM the observation that the mean relative abundance and variance are not independent implies that the mean is proportional to the carrying capacity $\bar{x}_i \propto K_i$, so evidence of a lognormal MAD informs us of the distribution of parameters used in the SLM to describe a community, not the SLM itself.

In order to assess the effect of migration on mean abundances, we compared the lognormal location and scale parameters (mean and standard deviation of the log of mean abundance) between transfer 12 (end of migration treatment) and transfer 18 (the last transfer in the experiment). Fig 2d shows that parameter combinations approached similar values as the experiment progressed, meaning that the shape of the MAD consistently converged to a similar form regardless of the migration treatment. We briefly note how a power law is unlikely to explain the MAD. While the lognormal can resemble a power law when the scale parameter is large, the fitted lognormal did not appear linear on a log-log scale (Fig 2c) and the scale parameter continued to decrease with time across experimental conditions [73]. In addition, inspecting the treatment-transfer combination with the most ASVs (regional migration, transfer 12), we see that the empirical MAD is not linear on a log-log scale, ruling out the appropriateness of a power law (S2 Fig).

## Testing the macroecological effects of migration

Our analysis has shown that demographic manipulations of experimental communities do not induce *qualitative* changes in macroecological patterns (e.g., altering the form of Taylor's Law). This result was consistent with the observation that natural microbial communities across disparate environments displayed similar patterns [21,32]. However, we have not yet fully examined the *quantitative* effects of these experimental manipulations. To determine whether these effects exist and the degree that they can be predicted, we examined deviations in macroecological quantities, incorporated experiment-specific forms of migration into the SLM, and tested its predictive capacity.

The previous section demonstrates that the SLM is a useful descriptive model of experimental microbial communities. Therefore, it is reasonable to expect that a form of the SLM that incorporates migration could serve as a predictive model. While a constant rate of migration appears intuitive and can be incorporated into the SLM (S4 Text), this model does not reflect the details of the experiment. Rather, in this experiment migrants only entered communities at the start of a given transfer cycle (Fig 3a). This detail corresponds to a model where the effect of migration is modeled as an experimentally-induced perturbation on the initial abundance of an ASV within a transfer cycle ($n_i^{(k)}(0)$; Materials and Methods, S6 Text). The full time-dependent solution of the SLM which depends on initial conditions has previously been solved [68], allowing one to model the temporal evolution of the AFD in response to experimental perturbations that alter initial abundances and compare their effects to AFDs with a constant rate of migration (S5 Text and S3 Fig). However, the time-dependent distribution of abundances is rather unwieldy, which lead us to numerically simulate the SLM.

In order to simulate the SLM it is necessary to identify the number of variables (i.e., community richness) and parameter values (i.e., carrying capacities). Using rarefaction curves and previously established richness estimation procedures [74], we found that the richness of the progenitor community was $\sim 100$ fold greater than the richness of a typical assembled community (S4 Fig). Migration had little effect on richness, as estimates were only slightly higher

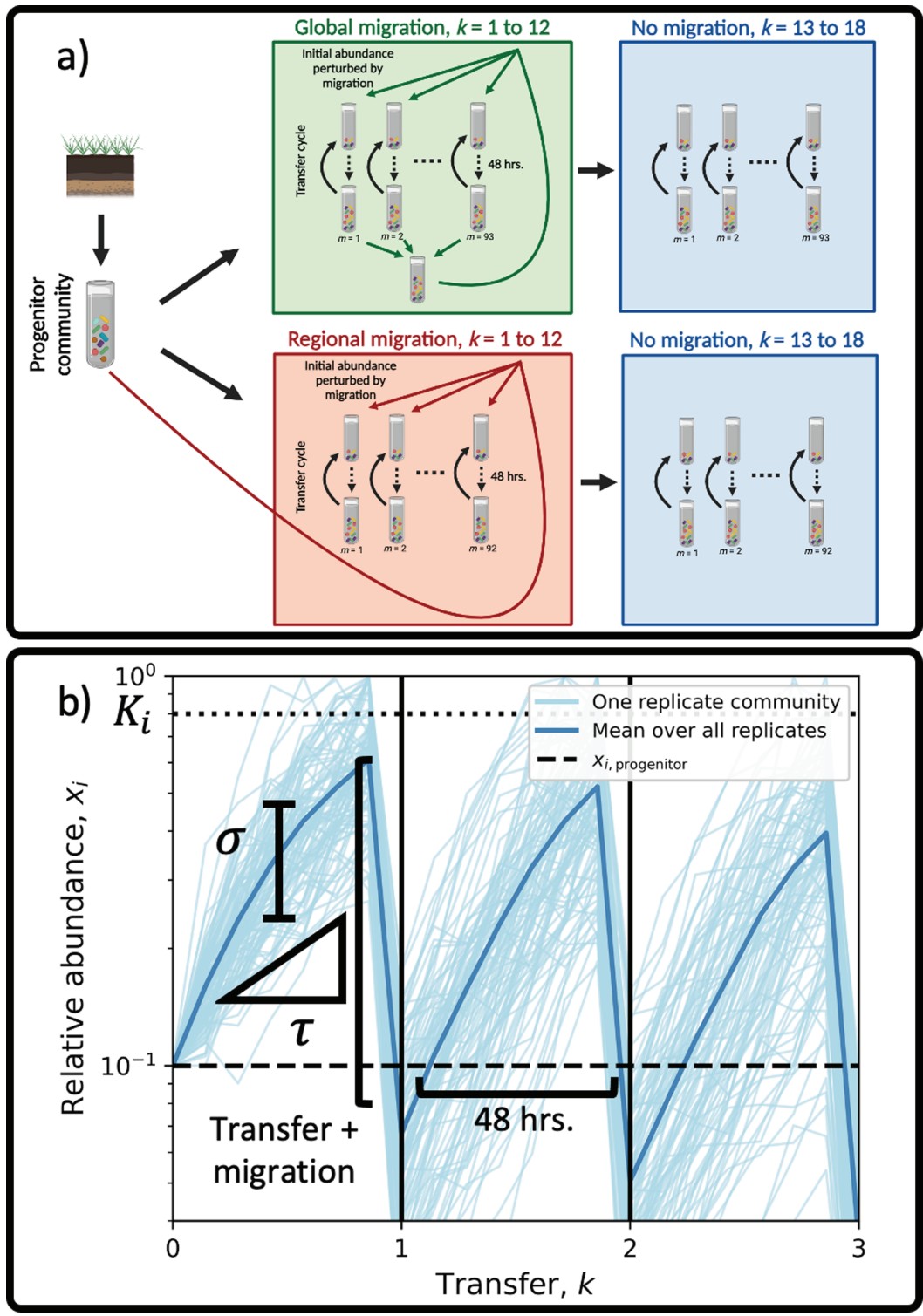

**Fig 3. Incorporating experimental details into the Stochastic Logistic Model. a)** Replicate communities were initiated from a single progenitor community that was isolated from soil. Communities were grown in microcosms containing a single carbon source for 48 hours, where they were then transferred to a microcosm with replenished resourced. This process constitutes a single transfer cycle. Migration was manipulated by altering the abundances of ASVs at the start of a transfer cycle. Two forms of migration were performed: regional and global. Regional migration represents a form of island-mainland migration, where aliquots of the progenitor community were added at the start of a transfer cycle. Global

migration was manipulated by mixing aliquots of each replicate community at the end of a transfer cycle, which was then redistributed at the start of the subsequent transfer cycle. In both cases migration manipulations were performed for the first 12 transfers (1 to 12) and ceased for the remaining six transfers (13 to 18). Community assembly experiments were also performed with no migration manipulations for the entirety of the 18 transfers (not pictured). **b)** We modeled the ecological dynamics of each ASV within a given transfer cycle by incorporating the forms of migration performed in the experiment into the Stochastic Logistic Model of growth (S6 Text). For a given migration treatment we inferred two parameters of the SLM: the strength of environmental noise ($\sigma$; represented as variation in abundance trajectories within a transfer cycle) and the timescale of growth ($\tau$; inverse of the maximum rate of growth). The parameter $\tau$ is particularly relevant for serial-dilution community assembly experiments, as it determines whether the community member reached its carrying capacity before the start of the subsequent transfer cycle. Migration altered the relative abundance of an ASV at the start of a transfer cycle as a perturbation of initial conditions (Eq 11). This experimental detail reduces the relative abundance of a given ASV if the relative abundance in a migration inoculum was lower than the carrying capacity ($K_i$), where the ASV then proceeds to increase in abundance at a rate set by the timescale of growth $\tau$. Conceptual diagram **a** was modified from Estrela et al. [53]. Fig 3a created with Biorender (ID j67r579)

among communities that experienced migration while remaining orders of magnitude lower than the richness of the progenitor community. This result suggests that the carrying capacity of a given ASV played a primary role in determining its survival, raising the question of how to 1) specify the carrying capacity of an ASV in the assembled communities ($K_i$) and 2) determine how the $K_i$ of an ASV relates to its abundance in the progenitor community. For the first question, the observation that Taylor's Law holds in the no migration treatments implies that the coefficient of variation of growth fluctuations was constant across ASVs (i.e., $\sigma_i \approx \sigma$), meaning that the mean relative abundance of an ASV across communities is proportional to its carrying capacity ($\bar{x}_i \propto K_i$). Given that $\bar{x}_i$ follows a lognormal distribution, $K_i$ should also follow a lognormal.

For the question of dependency between the carrying capacity of an ASV and its progenitor abundance, we found that distributions of $x_{i,\text{progenitor}}$ differed depending on whether a given ASV was present in the assembled communities, with ASVs that were present in assembled communities having a higher relative abundance in the progenitor (S5 Fig–S7 Fig). This statistical dependence between the progenitor and assembled communities, along with details related to sampling (i.e., number of reads; S8 Fig), were incorporated into our numerical SLM simulations (Materials and Methods, S6 Text) from which the strength of environmental noise ($\sigma$) and the timescale of growth ($\tau$) was inferred (S8 Text).

### Experiment-agnostic macroecological patterns

We first examined the quantitative effect of migration on the macroecological patterns that have been unified by the SLM: the AFD and Taylor's Law. Fig 4a–4c shows that the AFDs of ASVs rescaled by their mean and variance differed between transfers 12 (migration present) and 18 (migration absent). This shift was particularly notable in the regional migration treatment, whereas the global migration treatment AFDs appeared to be the most similar.

To examine the shift in the AFDs we calculated the KS distance between AFDs from transfer 18 and 12 for each ASV, focusing on ASVs present in every replicate community, and then calculated the mean KS over ASVs. Using Approximate Bayesian Computation (ABC), we identified the $\{\tau, \sigma\}$ pair that best explained the observed mean KS statistic (Materials and Methods; S6 Text). We then used the selected set of parameters to generate distributions of mean KS using $10^3$ SLM simulations. The discrepancy between the data and our predictions for the no migration treatment could be explained by the existence of multiple attractors, a scenario where certain ASVs had high abundances in a subset of communities and low

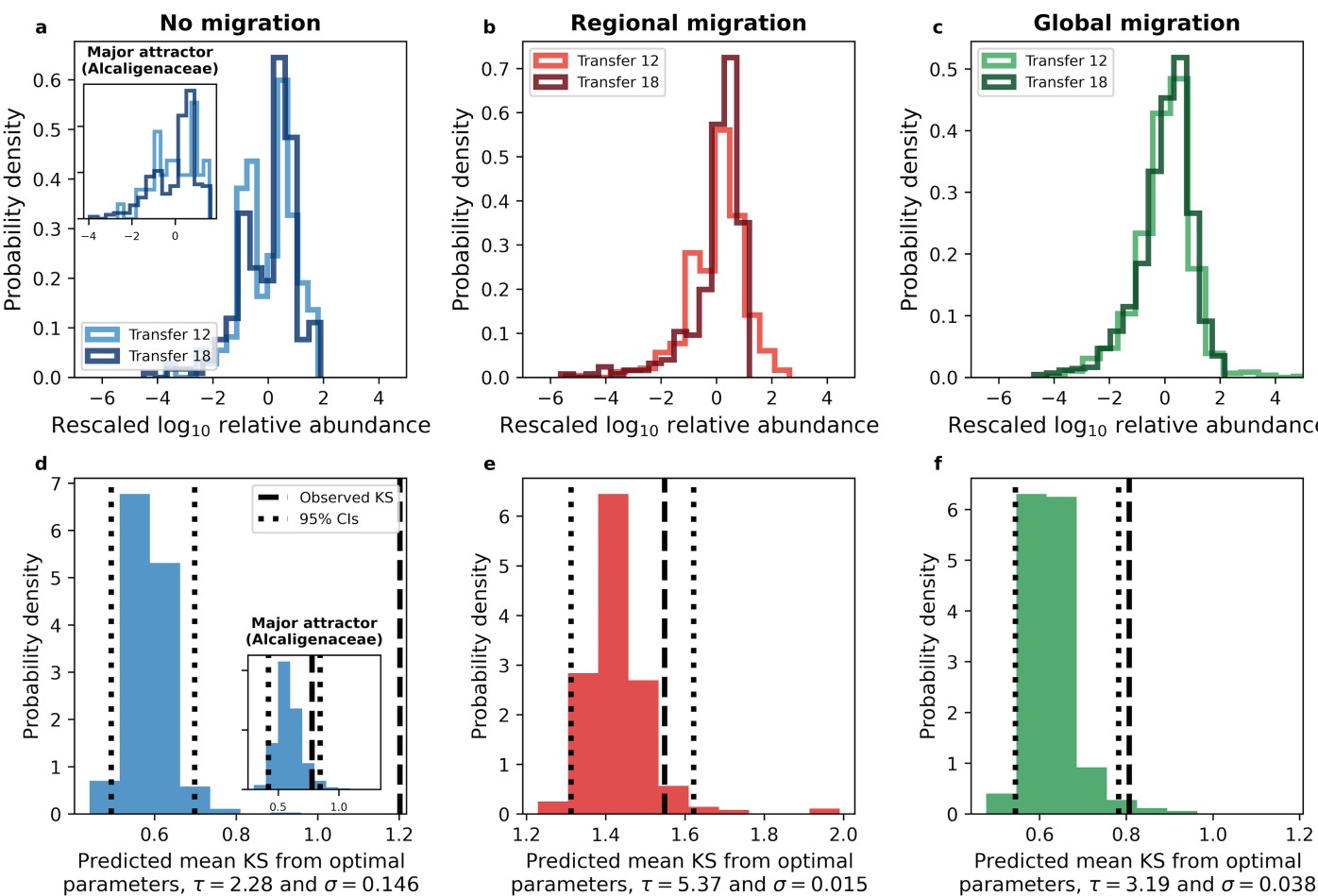

**Fig 4. Migration impacts the shape of AFDs. a-c)** By rescaling $\log_{10}$ transformed AFDs for each ASV, we examined how the shape of the AFD changed before and after the cessation of migration. We quantified this shift by estimating the KS distance between ASVs before (transfer 12) and after (transfer 18) the cessation of the migration treatment for each ASV, then calculated the mean over ASVs. **d-f)** Using the selected parameter combinations of the strength of environmental noise ($\sigma$) and the typical timescale of growth ($\tau$) identified by ABC, we found that the SLM predicted reasonable KS statistics for regional migration as well as within the major attractor of the no migration treatment (inset of sub-plots **a,d**), with borderline successful predictions for the global migration treatment. The solid vertical line represents the mean KS statistic from empirical data while the dashed vertical lines represent the 95% confidence intervals of the distribution of mean KS statistics calculated from our simulated data.

abundances in others, a qualitative deviation from the SLM [53]. Focusing on communities which belonged to the major attractor ($\sim 70\%$ of communities, S2 Table), we found that the SLM could predict the observed mean KS (insert of Fig 4d). The effect of regional migration was also captured by the SLM (Fig 4e), with the observed statistic for global migration communities lying slightly outside predicted range (Fig 4f). However, manipulating the SLM parameters revealed that the AFD exhibited the largest systematic deviation via migration for the regional migration treatment (S9 Fig). This result suggests that the inability of the SLM to fully capture the change in AFDs for the global migration treatment was due to its invariance to this particular form of migration.

We then analyzed how migration impacted the exponent of Taylor's Law (Fig 5a–5f). Under regional migration, we found that the exponent was lower than that of the no migration treatment at transfer 12 (Fig 5a and 5b), but approached the result of the no migration treatment by transfer 18 (Fig 5d and 5e). Similarly, the intercept of regional

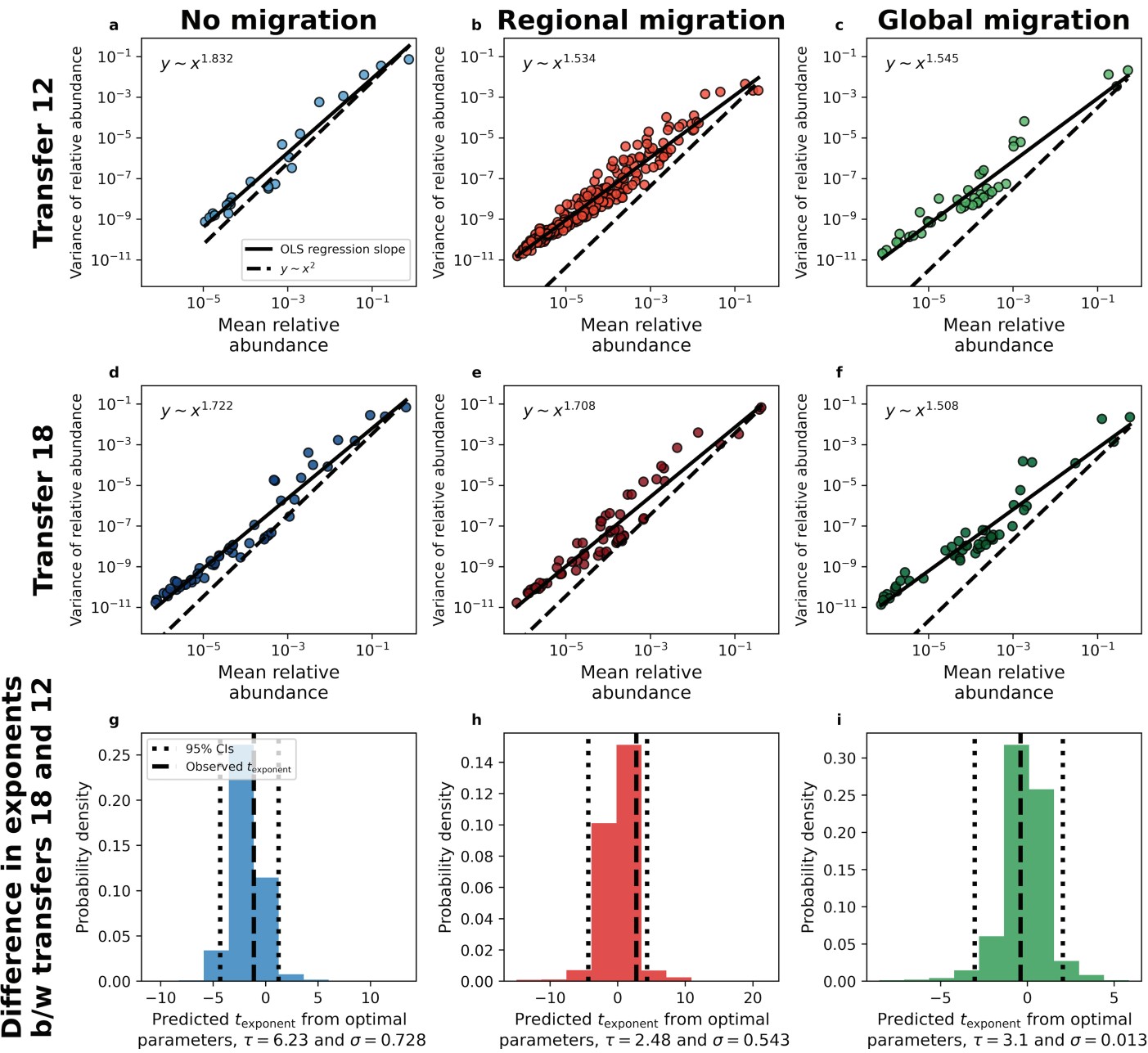

**Fig 5. Regional migration impacts the exponent of Taylor's Law. a-f)** By examining the exponent of Taylor's Law for each treatment we found that the exponent only considerably changed after the cessation of migration for the regional migration treatment. Each data point represents statistical moments calculated across replicate communities for a single ASV from a single experimental treatment. **g-i)** Using ABC-selected parameters (strength of environmental noise ($\sigma$) and the typical timescale of growth ($\tau$)) we found that the SLM succeeded in predicting the change in the exponent for regional migration.

migration treatment was initially higher, but proceeded to approach the value of the no migration treatment at treatment 18. In contrast, there was little change in the values of the exponent or intercept for the global migration treatment. In line with our predictions, there was no significant change in the exponent ($t_{exponent} = -1.1$, $P = 0.2$) or the intercept ($t_{intercept} = -0.6$, $P = 0.6$) between transfers 12 and 18 for the communities that did *not* undergo

migration. We found that global migration did not alter the exponent ($t_{\text{exponent}}$ = −0.4, $P$ = 0.7) or the intercept ($t_{\text{intercept}}$ = −0.4, $P$ = 0.8). Contrastingly, regional migration significantly altered both the exponent ($t_{\text{exponent}}$ = 3.3, $P$ = 0.02) and the intercept ($t_{\text{intercept}}$ = 2.75, $P$ = 0.03).

By repeating the same simulation procedure outlined for the AFD we found that the distribution of exponents generated could predict the observed values of $t_{\text{exponent}}$ across all migration treatments (Fig 5d–5f). However, by examining $t_{\text{exponent}}$ and $t_{\text{intercept}}$ across a grid of parameter combinations we found that the statistic was again only informative for the regional migration treatment (S10 Fig and S11 Fig). Our results demonstrate that the AFD and Taylor's Law were primarily informative of the effects of regional (i.e., island-mainland) migration.

### Experiment-specific macroecological patterns

The macroecological patterns we have examined up to this point can be explained by the SLM, though the AFD was not substantially altered by migration. It is useful to supplement our analyses by considering novel macroecological patterns that are likely to be altered by migration. Given the details of the experiment, we predicted that regional migration would alter typical abundances, whereas global migration would alter fluctuations in abundance (Materials and Methods).

**Regional migration.** Given the difference in abundance between assembled communities and the progenitor, we predicted that regional migration would alter the mean abundance of an ASV. We examined paired MADs for the regional and no migration treatments *before* ($k$ = 12) and *after* ($k$ = 18) the cessation of migration. The correlation between MADs was initially low and non-significant at transfer 12 but significantly increased by transfer 18 (Fig 6a and 6b; $Z_\rho$ = 2.7, $P$ = 0.01) [69]. Our SLM simulations recapitulated this observation, as we were able to predict the observed value for the selected parameter combination of $\sigma$ and $\tau$ from ABC (Fig 6c) and over a specific range of parameter combinations (S12 Fig). These results are consistent with the interpretation that ASVs reverted to their previous typical abundance (captured by the carrying capacity) once island-mainland migration ceased.

We then sought to determine whether the effect of regional migration depended on the abundance of an ASV in the progenitor community. To examine this dependence, we studied the relationship between the ratio of the mean abundance of an ASV in the regional and no migration treatments $\frac{\langle x_i \rangle_{\text{regional}}}{\langle x_i \rangle_{\text{no mig}}}$ and its abundance in the progenitor community. We found that at transfer 12 there was a strong relationship between the two quantities (statistically significant via permutation). By transfer 18 the slope was statistically indistinguishable from zero (Fig 6d and 6e). The observed change in slopes was significant ($t$ = −2.6, $P$ = 0.03) and could be reproduced using the SLM for certain $\sigma, \tau$ parameter combinations. These combinations overlapped with those that reproduced observed estimates of $Z_\rho$ (Fig 6f and S12 Fig), providing further validation of the SLM.

**Global migration.** According to the prediction of the model, global migration should strictly alter fluctuations in ASV abundance across replicate communities while leaving mean abundance unchanged (Materials and Methods; S7 Text). Specifically, the correlation in the MAD between global and no migration treatments should remain unchanged before and after the cessation of migration manipulations. This prediction held, as there was no significant change in the correlation between transfers 12 and 18 (S13 and S14 Fig; $Z_\rho$ = 0.2, $P$ = 0.4). However, there was also no significant increase in the strength of correlation for the distribution of CVs ($Z_\rho$ = 0.3, $P$ = 0.6), meaning that the cessation of migration did not considerably alter fluctuations in abundance.

# Regional migration statistics

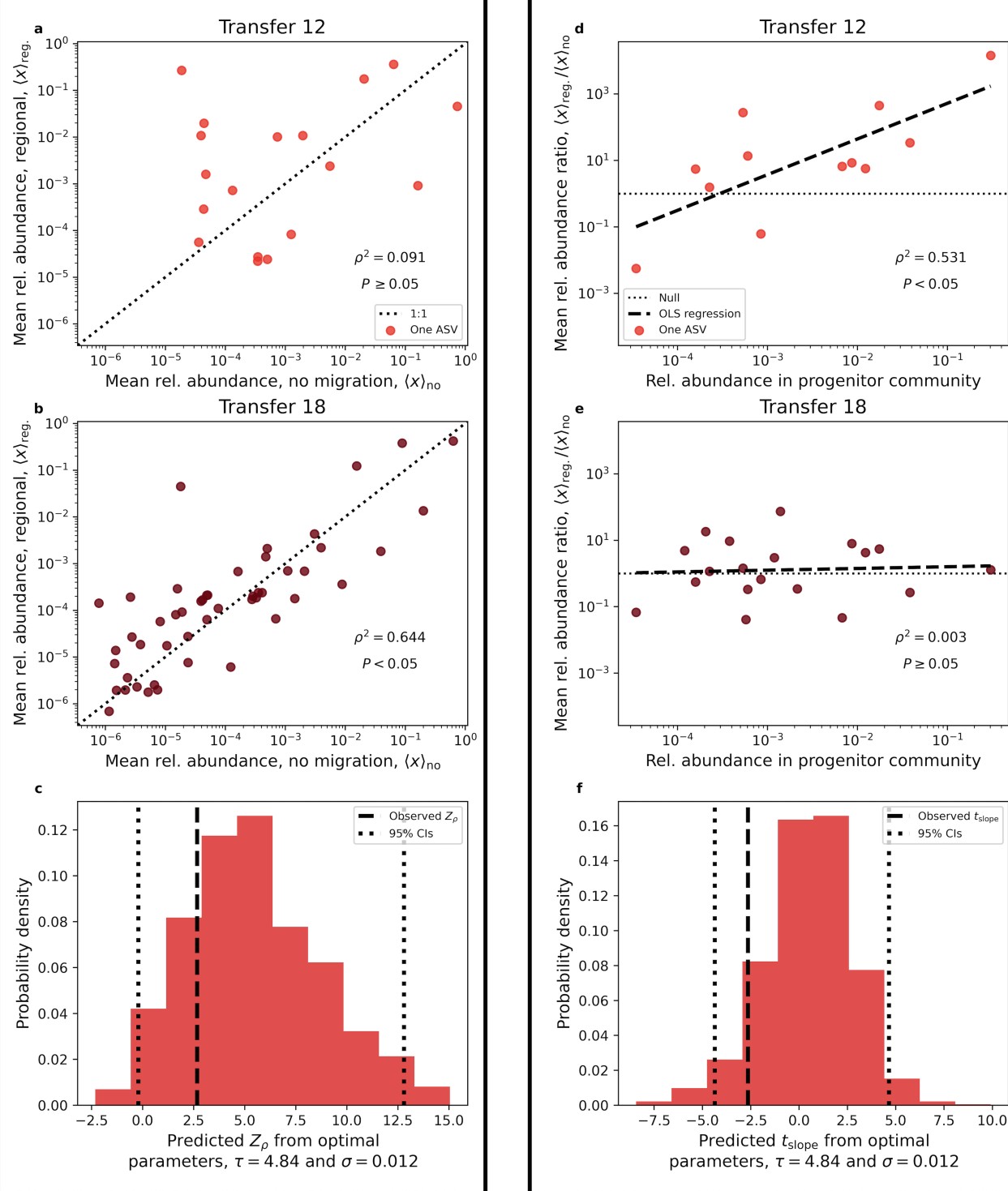

**Fig 6. Regional migration alters typical abundance in a manner consistent with SLM simulations.** The properties of the MAD were examined to evaluate the impact of migration over time. **a)** At transfer 12, the last transfer with migration, we found no apparent correlation between the MADs of the regional and no migration treatments. **b)** Contrastingly, the strength of the correlation rapidly increased by transfer 18. The significance of

this difference can be evaluated by calculating Fisher's $Z$-statistic, a statistic that can be applied to correlations calculated from simulated data. **c)** By performing simulations over a range of environmental noise strengths ($\sigma$) and timescales of growth ($\tau$), we visualized how the observed value of $Z_\rho$ compared to simulated distributions and identified reasonable parameter regimes as well as the relative error of our predictions. **d, e)** The ratio of the MAD between regional and no migration provided a single variable that could be compared to the abundance of an ASV in the progenitor community. There was a positive significant slope at transfer 12 that dissipated by transfer 18, reflecting the cessation of migration. Only sub-plots **c** and **f** contain simulated data.

It is possible that two time points are insufficient to detect the impact of global migration, as detecting shifts in fluctuations often require more observations than detecting shifts in typical values. As a solution, we leveraged the higher sampling resolution of the global migration treatment to examine the change in abundance between time points ($\Delta\ell$), as a subset of global migration communities were sequenced at each of the 18 transfers (S1 Table). The mean change in abundance across replicate communities at transfer $k$ ($\langle\Delta\ell^{(k)}\rangle$) tended to relax towards a value of zero around the sixth transfer and remained there throughout the rest of the experiment (S15 Fig). This trend indicates that the abundances of ASVs reached stationarity with respect to the start of the experiment by transfer 6, allowing us to examine equal intervals of transfer cycles before (7–12) and after (13–18) the cessation of migration.

Using permutational $t$-tests we determined whether $\langle\Delta\ell^{(k)}\rangle$ was altered after the cessation of migration, controlling for ASV identity. There was no evidence that $\langle\Delta\ell^{(k)}\rangle$ changes in either the no migration treatment ($\bar{t}$ = 0.1, $P$ = 0.3) nor in the global migration treatment ($\bar{t}$ = 0.09, $P$ = 0.1). This result is consistent with that prediction that global migration would not alter typical abundances.

Turning to fluctuations, we examined the CV of $\Delta\ell^{(k)}$. As predicted, the distribution of $CV^{(k)}_{\Delta\ell}$ did not increase for the no migration treatment ($\bar{t}$ = 0.09, $P$ = 0.8; Fig 7a). In the global migration treatment, the coefficient of variation $CV^{(k)}_{\Delta\ell}$ significantly increased after the cessation of migration ($\bar{t}$ = 1.3, $P<10^{-3}$; Fig 7b). This result is consistent with our prediction that global migration dampens fluctuations across communities. However, while our SLM simulations succeeded in predicting $\bar{t}$ for the no migration treatment, they failed to capture the effects of the global migration treatment (Fig 7c and 7d).

Experimental details may explain why the effect of the global migration treatment, while significant, was not particularly large, as the size of the inoculum was nearly two orders of magnitude smaller than that of regional migration (Table 1). However, this experimentally-imposed parameter does not explain why the observed value of $CV^{(k)}_{\Delta\ell}$ lay outside the predictions of the SLM. Attractor status also does not explain this discrepancy, as the global migration communities were previously classified as belonging to the same attractor (S2 Table, [53]). Instead of evaluating fluctuations *across* communities, we focused on fluctuations over time *within* individual communities to evaluate the effect, if any, of global migration. We found that estimates of the CV before ($CV^{<}_{\Delta\ell}$) and after the cessation of migration ($CV^{>}_{\Delta\ell}$) were similar for both the no and global migration treatments (Fig 7e and 7f). Using a measure of the change in CVs, we found that distributions of both treatments highly overlapped, meaning that the change in the CV *within a given community* did not considerably increase in the global migration treatment (Fig 7g). Splitting said distribution into high occupancy ASVs reinforced this conclusion, as there was no systematic increase in the change in CV for the global migration treatment (Fig 7h).

## Global migration statistics

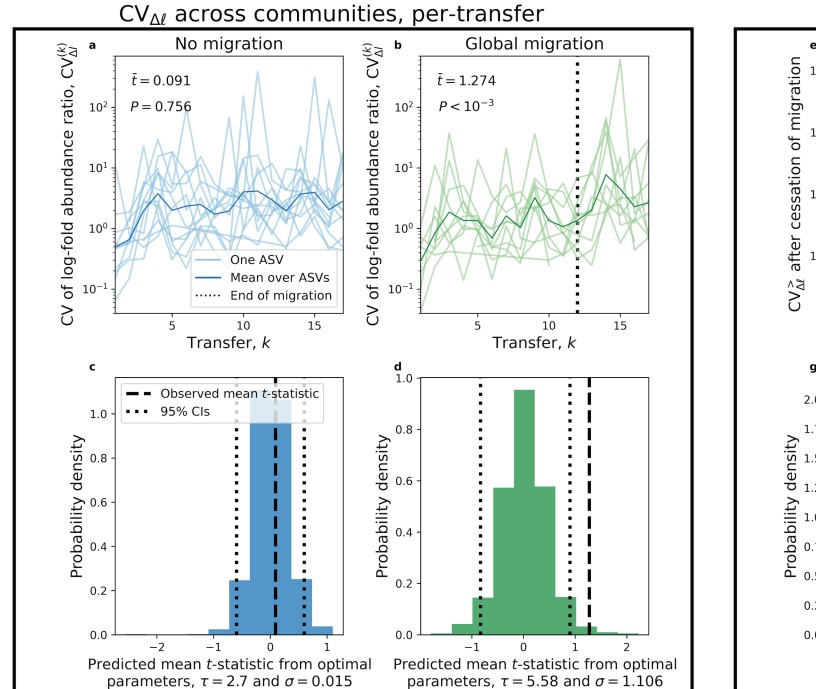

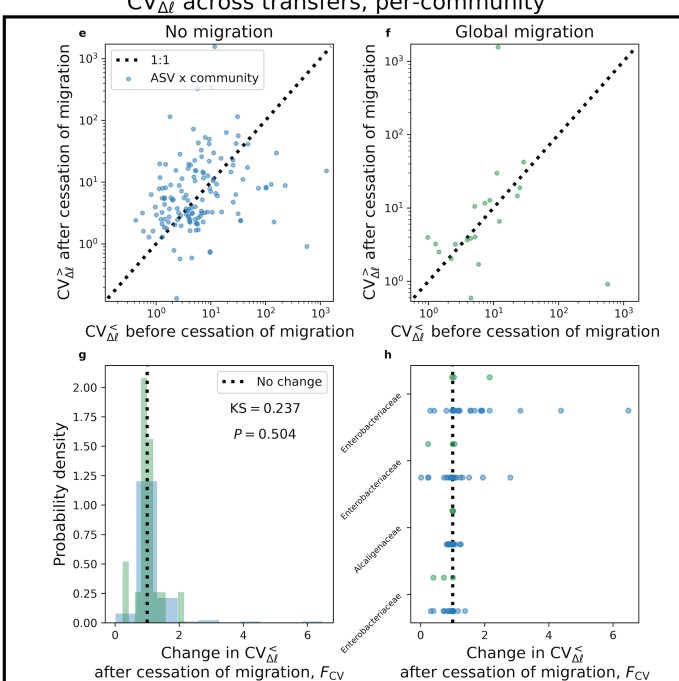

**Fig 7. SLM simulations are unable to reproduce log-fold fluctuations *across* communities under global migration.** By leveraging the entire timeseries we can examine how the coefficient of variation of $\Delta\ell$ was altered by migration for two scenarios: 1) fluctuations across replicate communities at a given transfer ($CV_{\Delta\ell}^{(k)}$) and 2) fluctuations before ($CV_{\Delta\ell}^{<}$) and after ($CV_{\Delta\ell}^{>}$) the cessation of migration within a given community. **a)** As expected, there was no change in $CV_{\Delta\ell}^{(k)}$ for the no migration treatment. **b)** However, $CV_{\Delta\ell}^{(k)}$ tended to slightly increase after transfer 12 for global migration, a shift that was found to be significant using a $t$-test where transfer labels were permuted for each ASV. **c, d)** Our SLM predictions using the ABC-selected strength of environmental noise ($\sigma$) and typical timescale of growth ($\tau$) succeeded for the no migration treatment, but was unable to capture observed values of $\bar{t}$ under global migration. These results pertaining to the fluctuations *across* communities can be contrasted with fluctuations within a community. **e, f)** We did not observe a systematic shift in the CV before vs. after the cessation of migration for either treatment. **g)** By calculating the difference between two CVs ($F_{CV}$) for each ASV in each replicate community, we did not observe a significant difference between migration treatments using a KS test constrained on ASV identity. **h)** This conclusion is confirmed by investigating the change in the CV for high occupancy ASVs, as there was no systematic difference between migration treatments.

## Discussion

The main result of this study is the demonstration that experimental microbial communities grown on a single carbon source (i.e., glucose) can sustain orders of magnitude of variation in abundance across community members. This variation provides the means to assess distributions of typical abundances across several orders of magnitude, a prerequisite for examining broad probabilistic patterns of diversity and using a macroecological approach. Meeting this criterion allowed us to document the existence of macroecological patterns that have only been documented in naturally occurring communities, suggesting a quantitative equivalence between experimental and observational studies despite the controlled nature of artificially maintained communities. However, the repeatable maintenance of variation observed in the face of demographic manipulations was likely contingent on the high level of variation present in the progenitor community. This figurative raw material is analogous to the need for genetic variation to exist before selection can occur [75–77], the absence of which would preclude the possibility of macroecological investigations.

Characterizing robust empirical patterns in artificial communities is a key step toward identifying predictive ecological models. The existence of patterns predicted by the SLM in artificial communities provided an opportunity to evaluate the macroecological consequences of experimental manipulations. Our approach of extending the SLM, an empirically validated model of microbial community composition, using experimental details proved to be a useful framework for identifying treatments that were capable of generating macroecological effects. We examined the effects of two different forms of migration: regional (island-mainland) and global (fully interconnected metacommunity) [53]. As expected, regional migration altered macroecological patterns of typical relative abundance in a manner that was captured by the SLM using minimal free parameters. The results of our regional migration analyses demonstrate that the SLM can link experimental communities and macroecological patterns.

Regarding global migration, we predicted that the treatment would primarily alter ASV fluctuations around their typical abundance [59]. We observed no change in the abundance fluctuations of community members after the cessation of migration *within individual communities*. However, we found that variation *across communities* tended to increase after the cessation of migration. This trend is consistent with our hypothesis regarding the effect of global migration, though it could not be reproduced by our model. This inconstancy between fluctuations *across* and *within* communities suggests that experimental communities readily reach a state where their statistical properties do not change over transfer cycles, but can vary from community-to-community. In physics parlance, experimental communities are stationary, but may not necessarily be ergodic [78]. The existence of alternative stable-states are a likely explanation, as they are common in experimental microbial communities [56,79,80], were present in the "no migration" treatment for our experiment of interest [53], and can be induced via fluctuations intrinsic to serial dilution experiments [81]. In addition, the effects of alternative stable-states have been used to explain invasion dynamics in experimental microbial communities [82]. The SLM as presented in this manuscript cannot capture the existence of alternative stable-states, as the abundance of each species fluctuate independently around the same value. While the SLM as-is cannot account for alternative stable-states, it has been previously modified to investigate the macroecological consequences of alternative stable states in natural microbial communities, but required temporal sampling effort which exceeds that of the experiment we considered [36]. Regardless, we can make two claims about the impact of alternative stable-states. First, the quantity of migrants chosen for the global migration treatment was sufficient to alter attractor status, but insufficient to alter fluctuations in abundance within a given community. Second, macroecological patterns captured by the SLM persisted despite the pervasiveness of alternative stable-states, implying that heterogeneous outcomes of community assembly do not qualitatively alter macroecological patterns.

In this study we focused on patterns relating to the typical abundance and fluctuations in abundance across communities as well as over time. Noticeably, we did not investigate patterns that likely require the explicit invocation of interactions between community members (e.g., the correlations of abundance fluctuations), as the addition of interaction terms into the SLM would require several assumptions about the network of interactions and their magnitude. However, the absence of interaction terms in the SLM does not mean that interactions did not contribute to the macroecological patterns we observed. Models such as the SLM can be viewed as "mean-field" models, where interactions between community members are permitted, but their effects on the dynamics of a given community member are determined by the mean dynamics of all community members [83]. Specifically, the SLM as

presented in this study will continue to hold if the effect of community interactions is restricted to the parameters of the SLM (e.g., carrying capacity). Investigating correlations in abundance between community members, which can be understood as an outcome of specific ecological interactions rather than the mean effect of the community, likely requires models that go beyond phenomenology by explicitly considering mechanisms such as resource consumption [12,84]. Indeed, consideration of resource consumption has proven critical for investigating the evolutionary dynamics of microorganisms in an ecological context [22].

Recent developments on the predictability of community function (e.g., total biomass, polysaccharide hydrolysis, resource excretion, etc.) point towards new avenues of exploration for microbial macroecology. There is increasing evidence that the functional profiles of experimental communities tend to follow quantitative rules that are amenable to mathematical modeling [85–88]. Extending studies of microbial macroecology beyond patterns of abundance and community composition to the level of function would fully realize the physiological and energetic breadth that allowed macroecology vital to advance our understanding of macrobial life [4].

It is worth discussing how the experimental implementation of migration relates to natural environments. Microbial community assembly experiments often examine ecological dynamics as a "boom-and-bust" phenomenon, where abundances are initially low and proceed to reach a carrying capacity. This decision is often made due to the unwieldy nature of managing an array of continuous cultures (i.e., chemostats). However, this experimental design may reflect pulse-like forms of migration in nature, as boom-and-bust dynamics can be found across diverse ecosystems. Examples of environments where boom-and-bust dynamics occur include pitcher plants [89], particles of organic matter in the open ocean [90,91], and even the human gut [92,93]. Boom-and-bust dynamics can serve as a useful model for environments where resources are periodically supplied, as migrants must persist until the environment becomes favorable to growth (e.g., surviving in a metabolically inactive state [94–96]). Migration in such systems is difficult to quantify, though recent evidence suggests that it is sufficiently high such that microbes are rarely endemic to the environment in which they are found [97]. One could then predict that the shift in the AFD and the exponent of Taylor's Law observed in this experiment may hold for natural communities. However, a key parameter of our modeling efforts was the number of migrants transferred at the start of a given transfer cycle relative to the size of the community. Here the value of that parameter was known because it was chosen by the experimenters. In natural systems that parameter is rarely known, though one could estimate it by tracking the rate that cells enter and exit the system. If such measurements were obtained for a variety of natural systems, one could theoretically investigate the dependency of macroecological patterns on migration rates in natural microbial communities.

Finally, we consider how the results presented here shape the microbial view of macroecology. The discipline of macroecology was originally conceived as an explicitly non-experimental form of investigation [4]. Analysis of the origin and development of macroecology provides two historical explanations for the initial rejection of experimental approaches: 1) large-scale community-level experiments were often impractical and 2) producing generalities from experiments has proven to be difficult [98]. Our results demonstrate that these two constraints are ameliorated by the features of microbial communities, providing counter evidence to recent claims that statistical distributions do not provide information about ecological mechanisms [99]. The timescales, abundance, and comparative ease with which ensembles of communities can be maintained and manipulated make microorganisms an ideal system for testing quantitative macroecological predictions.

## Supporting information

**S1 Text. Additional experimental details.** Additional information about the community assembly experiment.
(PDF)

**S2 Text. Error estimates.** Definition of relative error.
(PDF)

**S3 Text. Phenomenological logistic growth from a consumer-resource model.** Derivation of logistic growth in a batch culture experimental setup.
(PDF)

**S4 Text. Deriving the stationary AFD for the SLM with a constant rate of migration.** Derivation of the stationary distribution of abundance for the SLM with a constant rate of migration.
(PDF)

**S5 Text. Obtaining the time-dependent AFD for the SLM.** The time-dependent probability distribution of abundance for the SLM.
(PDF)

**S6 Text. Simulating the SLM.** Additional details and derivations related to our SLM simulations.
(PDF)

**S7 Text. Effect of global migration on the statistical moments of abundance.** Derivations of the effect of global migration on moments of abundance in a batch culture experimental design framework.
(PDF)

**S8 Text. Inferring SLM parameters.** Additional details about SLM parameter inference.
(PDF)

**S1 Fig. The sampling form of the gamma distribution predicts ASV occupancy. a)** The fraction of replicates harboring a given ASV (i.e., occupancy) can be predicted using a form of the gamma distribution that accounts for sampling across experimental treatments and transfers. **b)** The distribution or relative errors exhibited a similar form across treatments and transfers, suggesting that the predictions of the SLM are broadly applicable. **c)** By comparing ASVs that were present in both migration and no migration treatments, we can see that the errors are generally similar between treatments, if only slightly higher in the no migration treatment.
(TIFF)

**S2 Fig. Lognormal MAD of regional migration treatment.** In Fig 2d the MAD of each treatment were separately rescaled to facilitate comparison, meaning that a single lognormal was fit. The lognormal parameters ($\mu$ and $s$) slightly vary from treatment-to-treatment, so lognormal fits were examined for separate treatments. Here we present the MAD with fitted lognormal for the regional migration treatment at transfer 12, the treatment with the highest number of ASVs. We see that the empirical MAD is non-linear on a log-log scale, suggesting that a power law would not serve as an appropriate descriptor.
(TIFF)

**S3 Fig. The effect of migration on the AFD when modeled as a perturbation of initial conditions vs. a constant rate.** We examined how migration as a perturbation of initial conditions compared to a commonly assumed form of migration where it occurs at a constant rate per-unit time. The AFD of a form of the SLM with a constant rate of migration at stationarity was derived (S4 Text) and the time-dependent solution of the SLM was obtained from a prior study (S5 Text) . The following parameters were used: $K_i = 10^{-3}$, $\sigma_i = 0.7$, $\tau = 1$, and $x_i(0) = m_i = 10^{-2}$. We have rescaled time using the timescale of growth to arrive at a dimensionless parameter $\frac{t}{\tau}$. The AFD with no experimentally-imposed migration is represented by Eq 2 (i.e., a gamma distribution).
(TIFF)

**S4 Fig. Rarefaction curves for each treatment.** Rarefaction curves demonstrate how the richness (# ASVs) is  100-fold lower in descendant communities relative to the progenitor.
(TIFF)

**S5 Fig. Progenitor abundance vs. assembled mean abundance.** There is effectively zero correlation between the relative abundance of an ASV in the progenitor community and its mean abundance among descendant communities for all treatments at both transfers 12 and 18.
(TIFF)

**S6 Fig. Presence in descendant as function of progenitor abundance. a)** An ASV is more likely to be present in the descendant communities if it has a higher relative abundance in the progenitor. **b)** This result implies that probability that an ASV has a non-zero carrying capacity is a function of its progenitor abundance, a relationship that can be modeled as a logistic regression.
(TIFF)

**S7 Fig. The distribution of abundances in the progenitor community is consistent across treatment status. a–d)** The abundances of ASVs in the progenitor that are present in a given treatment are consistently shifted to the right, implying that the probability of an ASV surviving is conditional on its initial abundance. Permutation-based two-sample Kolmogorov—Smirnov tests were performed for each treatment.
(TIFF)

**S8 Fig. The distribution of total read counts.** In our simulations the generation of reads from relative abundances was done as a multinomial sampling process, where the total number of reads of a given replicate at a given time point was drawn from the empirical distribution of total read counts.
(TIFF)

**S9 Fig. AFD simulations. a–c)** The KS distance (KS) between AFDs from transfers 12 and 18 for all migration treatments across a parameter grid. **d–f)** Using the empirical KS distance, we obtained the mean relative error for each simulation. We see that the error of our predictions systematically tends to cluster for certain parameter regimes, particularly for regional migration. This pattern suggests that there are adjacent parameter regimes where the SLM adequately performs. For each parameter combination 100 iterations were performed.
(TIFF)

**S10 Fig. Taylor's Law exponent simulations.** The equivalent analysis as shown in S9 Fig for the exponent of Taylor's Law.
(TIFF)

**S11 Fig. Taylor's Law intercept simulations.** The equivalent analysis as shown in S9 Fig for the intercept of Taylor's Law. Similar to S10 Fig the errors tend to cluster together for regional migration, implying that the SLM is performing adequately for adjacent combinations of parameter regimes.
(TIFF)

**S12 Fig. Simulations of regional migration statistics.** The equivalent analysis as shown in S9 Fig–S11 Fig for statistics that capture the directional change in abundance caused by regional migration. The parameter regimes with lowest error correspond to the regions with lowest error for our regional migration Taylor's Law analyses (S10 Fig and S11 Fig).
(TIFF)

**S13 Fig. Global migration does not alter the MAD nor the distribution of CVs.** Similarly to the regional migration patterns examined, we evaluated whether a change in correlation occurred for the global migration treatment for the paired MAD and distribution of CVs. Within each time point for each measure, the strength of the correlation of significantly greater than zero. However, the change in correlation $Z_\rho$ was not significant for both cases, a result that is consistent with simulation results.
(TIFF)

**S14 Fig. Simulations of global migration statistics.** The equivalent analysis as shown in S9 Fig for statistics that capture the change in the fluctuations around the typical abundance caused by global migration.
(TIFF)

**S15 Fig. Mean change in $\Delta\ell$ under global and no migration.** For both **a)** no and **b)** global migration the mean of $\Delta\ell$ is initially higher than the stationary value of zero, though the mean relaxes to zero by transfer six for both treatments and does not appear to change after the cessation of global migration. This result is consistent with predicted consequences of global migration.
(TIFF)

**S1 Table. The number of replicate communities sequenced for a given treatment at a given transfer.**
(PDF)

**S2 Table. Attractor status.** The percent of communities belonging to a given attractor for each migration treatment.
(PDF)

## Acknowledgments

We thank S. Estrela for her assistance in reprocessing the data. We thank S. Bubnovich, M. Dal Bello, A. Goyal and members of the qEcoEvo group at ICTP for helpful discussions. We thank B.H. Good, O. Mazzarisi, M. Sireci, and N. I. Wisnoski for their comments on the manuscript. Conceptual diagrams created in BioRender under a CC BY 4.0 license.

## Author contributions

**Conceptualization:** William R. Shoemaker, Álvaro Sánchez, Jacopo Grilli.

**Data curation:** William R. Shoemaker, Álvaro Sánchez.

**Formal analysis:** William R. Shoemaker, Álvaro Sánchez, Jacopo Grilli.

**Funding acquisition:** William R. Shoemaker, Álvaro Sánchez.

**Investigation:** William R. Shoemaker, Jacopo Grilli.

**Methodology:** William R. Shoemaker, Jacopo Grilli.

**Project administration:** Álvaro Sánchez, Jacopo Grilli.

**Resources:** William R. Shoemaker.

**Software:** William R. Shoemaker.

**Supervision:** Álvaro Sánchez, Jacopo Grilli.

**Validation:** William R. Shoemaker.

**Visualization:** William R. Shoemaker.

**Writing – original draft:** William R. Shoemaker, Álvaro Sánchez, Jacopo Grilli.

**Writing – review & editing:** William R. Shoemaker, Álvaro Sánchez, Jacopo Grilli.

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
