## [Decision Letter · Decision Letter 0]

20 Feb 2025

PCOMPBIOL-D-24-02000

Macroecological patterns in experimental microbial communities

PLOS Computational Biology

Dear Dr. Shoemaker,

Thank you for submitting your manuscript to PLOS Computational Biology. After careful consideration, we feel that it has merit but does not fully meet PLOS Computational Biology's publication criteria as it currently stands. Therefore, we invite you to submit a revised version of the manuscript that addresses the points raised during the review process.

Please submit your revised manuscript within 30 days Apr 21 2025 11:59PM. If you will need more time than this to complete your revisions, please reply to this message or contact the journal office at ploscompbiol@plos.org. Please include the following items when submitting your revised manuscript:

We look forward to receiving your revised manuscript.

Kind regards,

Nic Vega, Ph.D.

Academic Editor

PLOS Computational Biology

Tobias Bollenbach

Section Editor

PLOS Computational Biology

**Additional Editor Comments :**

The reviews were overall positive but indicated a need for revision of the structure and presentation of the manuscript itself to increase clarity and accessibility of the contents. The requested revisions are, however, essentially editorial in nature and should require little if any new work.

**Journal Requirements:**

At this stage, the following Authors/Authors require contributions: William R. Shoemaker, Alvaro Sanchez, and Jacopo Grilli. Please ensure that the full contributions of each author are acknowledged in the "Add/Edit/Remove Authors" section of our submission form.

5) We have noticed that you have uploaded Supporting Information files, but you have not included a list of legends. Please add a full list of legends for your Supporting Information files after the references list.

Potential Copyright Issues:

i) Figures 1A, and 3A. Please confirm whether you drew the images / clip-art within the figure panels by hand. If you did not draw the images, please provide (a) a link to the source of the images or icons and their license / terms of use; or (b) written permission from the copyright holder to publish the images or icons under our CC BY 4.0 license. Alternatively, you may replace the images with open source alternatives. See these open source resources you may use to replace images / clip-art:

7) Thank you for uploading your study's underlying data set. There is a GPL-3.0 license on your data. We would encourage you to consider using a license that is no more restrictive than CC BY, in line with PLOS’ recommendation on licensing (http://journals.plos.org/plosone/s/licenses-and-copyright). 

8) Please amend your detailed Financial Disclosure statement. This is published with the article. It must therefore be completed in full sentences and contain the exact wording you wish to be published.

1) State what role the funders took in the study. If the funders had no role in your study, please state: "The funders had no role in study design, data collection and analysis, decision to publish, or preparation of the manuscript.".

**Reviewers' comments:**

Reviewer's Responses to Questions

Reviewer #1: The manuscript investigates whether broader patterns observed in the ecology hold when examining experiments in laboratory. As the authors mention, this is an important step towards enabling relevant, causal studies using experimental ecology. For this, the manuscript studies patterns emerging from the stochastic logistic model (SLM) and uses high-replicate assembled communities of soil bacteria in a controlled experimental setting.

The manuscript is thorough and well-written, and it follows a logical progression. Necessary information regarding the motivations, the hypotheses, the model assumptions, and the experimental details are included, further strengthening the manuscript. Especially, I appreciate it that the authors have included the detailed derivations related to their patterns of interest in the Methods section. These derivations serve as an important point of reference, without disrupting the flow of the main text.

In my opinion, this manuscript is an important contribution to the field and will be a very beneficial stepping-stone to enable future work in this direction. I only have a few minor suggestions, listed below.

Minor comments

1. For Fig 2c, the lower trend of the predicted pattern is not captured in the experimental data. I think it would be helpful to discuss why that might be.

2. In Fig 2d, please include more description of the two free parameters. In its current form, it takes too much extra effort for the reader to find the relevant information (which is needed to interpret Fig 2d).

3. In Fig 4 caption, I recommend mentioning what the optimal parameters represent. The values are mentioned in the figure and the description is included in the text, but I think it would be helpful to just briefly mention that they are from fitting the model into data.

4. In Figures 4, 5, and 6, I suggest using a consistent range for the x axis (and the y axis) across different cases (e.g. no, regional, and global migration) to make comparisons easier.

Reviewer #2: In this manuscript, the authors present what appears to be the first analysis in the literature of macroecological laws applied to experimental microbial communities grown in laboratory conditions. These communities, isolated from soil, were cultivated in microcosms containing a single carbon source over a 48-hour period, with the experimental results previously published in an earlier study.

The manuscript focuses on two main objectives. The first is to verify the validity (or potential variation) of three macroecological laws recently proposed by the last author of this manuscript for microbial communities from diverse origins. Specifically, these laws include: (1) the abundance-fluctuation distribution of bacterial species follows a gamma distribution, (2) mean abundances and variances are not independent but are linked by Taylor's law, and (3) mean abundances are distributed according to a lognormal distribution.

The first conclusion is that these laws do qualitatively hold in in-lab grown communities with relatively good accuracy even if the curves are not as precise as in "wild" communities. On the other hand, migration (especially regional migration) can affect the shape of the abundance fluctuation distribution. The observation of this shift in AFD is attributed to the possibility of alternative states, where some species can have large abundances in some states and low in others, implying some sort of ergodicity breaking.

The first conclusion is that these laws qualitatively hold in laboratory communities with relatively good accuracy, although the curves are not as precise as those observed in "wild" communities. However, migration, particularly regional migration, can influence the shape of the abundance-fluctuation distribution (AFD). This shift in AFD is attributed to the possibility of alternative states, where certain species exhibit high abundances in some states and low abundances in others, suggesting a form of ergodicity breaking.

The second focus is to determine whether the stochastic logistic model (SLM), which, along with additional assumptions, has been used to justify the aforementioned macroecological laws, can be adapted to model experimental setups that incorporate two types of migration. In the first type, termed regional migration, new individuals from a source (mother) community are added after a serial dilution, assuming a typical mainland-island dynamic. In the second type, referred to as global migration, multiple communities evolve in parallel and exchange individuals during serial dilution steps. The main conclusion is that the SLM is able to qualitatively explain the observed patterns in both cases. However, some subtle differences arise in quantitative comparisons, particularly in the case of regional migration.

Overall, the paper is well-written, with careful statistical and mathematical analyses. The results are robust, and the conclusions are both relevant and meaningful. Therefore, I fully support the publication of this manuscript in PCB.

However, I encourage the authors to consider the following suggestions for potential improvements or clarifications.

- Could the authors elaborate on why it was necessary to redo the analyses of Estrela et al. using ASVs? Including a brief comment or discussion on the distinction between ASVs and OTUs would provide helpful context.

- The fit shown in Fig. 2, particularly the one to the gamma distribution, doesn't appear to be very good. Have the authors tested any alternative fits that might provide a better correction to the gamma distribution? Similarly, for the third law, while it seems marginally compatible with a lognormal distribution, it could also potentially follow a power law. Please comment on this.

- In recent work (Proceedings of the National Academy of Sciences, 120 (44), e2215832120) it was argued that demographic noise is the best proxy for describing the dynamics and steady-state distribution of microbial communities. Could the authors elaborate a little bit on this point and specify why they favor environmental noise in their current analyses?

- I find the Materials and Methods section quite lengthy and, at times, tedious. While I agree that all relevant details should be included, would it be possible to condense this section? Alternatively, secondary details could be moved to the appendices or supplementary information (SI).

- Could the authors make an additional effort to condense and distill their findings in the Result section, presenting them in a more straightforward manner? The reader needs to remain highly focused in order not to get lost in the variety of migration forms, experiment-agnostic versus experiment-specific patterns, and the various deviations (or lack thereof) from SLM predictions.

- I do not find the discussion about alternative stable states particularly clear. While I have no objections to the idea, I’m convinced the authors could present a stronger case to better illustrate the existence of such alternative states. In particular, I don't fully understand how the SLM, even when studied under the serial dilution and migration setup, can lead to a bimodal distribution of species abundances. Could you please make it more transparent for the readers?

- Similarly, do the authors provide a strong explanation for why the exponents of Taylor's law change depending on the type of experiment, as reported in Fig. 5? More broadly, can any meaningful interpretation be made of the slope value?

- When the authors state, "Our results demonstrate that the AFD and Taylor’s Law were primarily informative of the effects of regional (island-mainland) migration," do they imply that by empirically determining these patterns in real communities, one can draw conclusions about the type of migration dominating the community? Please clarify.

- The meaning of the vertical lines in Fig.4 is not specified in the figure caption

In summary, I greatly appreciate the research and the results, but I believe the authors could improve the presentation of their findings and structure the manuscript in a more straightforward manner.

Reviewer #3: In this study, the authors bridge macroecological patterns with experimental ecology by applying specific ecological theory models. The manuscript presents a compelling premise: that a model such as SML, when applied to microbial community experiments in microcosms, can reveal ecological patterns similar to those observed in nature and therefore bridge macroecology with experimental ecology.

While the article is generally well written, a gap emerges in the description of the previously published experiment from which all data were used. The experiment is not fully explained, requiring the reader to consult another article for essential details. Instead, key information—such as the general premise of the experiment, its design rationale, and the core concepts it was intended to test—should be incorporated into the introduction. Providing this context would give readers a clearer understanding of why this particular experiment is well-suited as a bridging study between macroecological patterns and experimental ecology.

Minor Comments:

Line 85 – You briefly describe the experiment in citation #1 and mention that all microcosms contained a single supplied carbon source. You then specify glucose (i.e., glucose), and Figure 1a also shows only glucose examples. Were all microcosms using glucose, or were different carbon sources included? Please clarify.

Line 87 – Can you specify the dilution ratio of the aliquot? Was it a 1:10 dilution, 1:100, or another ratio?

Line 107 – Did you sequence only the final transfer cycle of each microcosm? Please clarify.

Line 557 – What exactly do you mean by "large ecological variation"? Could you be more specific?

**Have the authors made all data and (if applicable) computational code underlying the findings in their manuscript fully available?**

Reviewer #1: Yes

Reviewer #2: Yes

Reviewer #3: Yes

PLOS authors have the option to publish the peer review history of their article (what does this mean?). If published, this will include your full peer review and any attached files.

Reviewer #1: No

Reviewer #2: No

Reviewer #3: No

**Figure resubmission:**
---

## [Editor Report · Decision Letter 1]

10 Apr 2025

Dear Dr Shoemaker,

We are pleased to inform you that your manuscript 'Macroecological patterns in experimental microbial communities' has been provisionally accepted for publication in PLOS Computational Biology.

Best regards,

Nic Vega, Ph.D.

Academic Editor

PLOS Computational Biology

Tobias Bollenbach

Section Editor

PLOS Computational Biology

---

## [Editor Report · Acceptance letter]

PCOMPBIOL-D-24-02000R1

Macroecological patterns in experimental microbial communities

Dear Dr Shoemaker,

I am pleased to inform you that your manuscript has been formally accepted for publication in PLOS Computational Biology. Your manuscript is now with our production department and you will be notified of the publication date in due course.

With kind regards,

Anita Estes
